# A Proximal Block Coordinate Descent Algorithm for Deep Neural Network Training

**Tim Tsz-Kit Lau**[*], **Jinshan Zeng**[†*], **Baoyuan Wu**[‡], **Yuan Yao**[*]
[*]Department of Mathematics, The Hong Kong University of Science and Technology
[†]School of Computer and Information Engineering, Jiangxi Normal University
[‡]Tencent AI Lab

## Abstract

Training deep neural networks (DNNs) efficiently is a challenge due to the associated highly nonconvex optimization. The backpropagation (backprop) algorithm has long been the most widely used algorithm for gradient computation of parameters of DNNs and is used along with gradient descent-type algorithms for this optimization task. Recent work have shown the efficiency of block coordinate descent (BCD) type methods empirically for training DNNs. In view of this, we propose a novel algorithm based on the BCD method for training DNNs and provide its global convergence results built upon the powerful framework of the Kurdyka-Łojasiewicz (KL) property. Numerical experiments on standard datasets demonstrate its competitive efficiency against standard optimizers with backprop.

## 1 Introduction

Backprop (Rumelhart et al., 1986) is the most prevalent approach of computing gradients in DNN training. It is mostly used together with gradient descent-type algorithms, notably the classical stochastic gradient descent (SGD) (Robbins & Monro, 1951) and its variants such as Adam (Kingma & Ba, 2015). Regardless of the optimizers chosen for network training, backprop suffers from vanishing gradients (Bengio et al., 1994; Pascanu et al., 2013; Goodfellow et al., 2016). Various methods were proposed to alleviate this issue, e.g., rectified linear units (ReLUs) (Nair & Hinton, 2010) and Long Short-Term Memory (Hochreiter & Schmidhuber, 1997), but these methods are unable to completely tackle this inherent problem to backprop. One viable alternative to backprop with gradient-based optimizers to avoid vanishing gradients is to adopt gradient-free methods, including (but not limited to) the alternating direction method of multipliers (ADMM) (Taylor et al., 2016; Zhang et al., 2016) and the block coordinate descent (BCD) method (Carreira-Perpiñán & Wang, 2014; Zhang & Brand, 2017). The main idea of ADMM and BCD is to decompose the highly coupled and composite DNN training objective into several loosely coupled and almost separable simple subproblems. The efficiency of both ADMM and BCD has been illustrated empirically in Taylor et al. (2016), Zhang et al. (2016) and Zhang & Brand (2017). Meanwhile, BCD has been tremendously studied for nonconvex problems in machine learning (see e.g., Jain & Kar, 2017).

In this paper, we propose a novel algorithm based on BCD of Gauss-Seidel type. We define the loss function using the quadratic penalty method (Nocedal & Wright, 1999) by unrolling the nested structure of DNNs into separate "single-layer" training tasks. This algorithm involves simplifications of commonly used activation functions as projections onto nonempty closed convex sets (commonly used in convex analysis (Rockafellar & Wets, 1998)) so that the overall loss function is block multiconvex (Xu & Yin, 2013). This property allows us to obtain global convergence guarantees under the framework of KL property (Attouch et al., 2013; Bolte et al., 2014).

## 2 Related work

Carreira-Perpiñán & Wang (2014) and Zhang & Brand (2017) also suggest the use of BCD for training DNNs and observe empirically the per epoch efficiency where the training loss drops much faster than SGD. Multiple related work consider a similar scheme to ours. A very recent piece of work (Frerix et al., 2018) implements proximal steps for model parameter updates only but keep

gradient steps for updating the activation parameters and the output layer parameters, whilst we also apply proximal steps for updating these parameters. In the problem formulation of Zhang & Brand (2017), bias vectors are not used in the layers. They concatenate all weight matrices in all hidden layers and all activation vectors (defined below) respectively into two separate blocks and update together with the weight matrix of the output layer so that these three blocks are updated alternately instead of an overall backward order as in backprop. Carreira-Perpiñán & Wang (2014) consider a specific DNN using squared loss and sigmoid activation function, and propose the so-called method of auxiliary coordinate (MAC). Our problem formulation is similar, but is further simplified described in the next section.

## 3 THE PROXIMAL BLOCK COORDINATE DESCENT ALGORITHM

### 3.1 PRELIMINARIES AND NOTATIONS

We consider a feedforward neural network with $L$ hidden layers. Let $d_\ell$ be the number of nodes of the $\ell$-th layer, $N$ be the number of training samples and $K$ be the number of classes. Note that $d_{L+1} = K$. We adopt the following notations: $\boldsymbol{x}_j \in \mathbb{R}^{d_0}$ the $j$-th training data, $\boldsymbol{X} = (\boldsymbol{x}_1, \dots, \boldsymbol{x}_N) \in \mathbb{R}^{d_0 \times N}$, $\boldsymbol{y}_j \in \mathbb{R}^{d_{L+1}}$ the one-hot vector of its corresponding label, $y_{ji}$ the $i$-th entry of the column vector $\boldsymbol{y}_j$, $\boldsymbol{Y} = (\boldsymbol{y}_1, \dots, \boldsymbol{y}_N) \in \mathbb{R}^{d_{L+1} \times N}$, $\boldsymbol{W}_\ell \in \mathbb{R}^{d_\ell \times d_{\ell-1}}$ the weight matrix between the $\ell$-th and $(\ell-1)$-th hidden layers, $\boldsymbol{b}_\ell \in \mathbb{R}^{d_\ell}$ the bias vector of the $\ell$-th hidden layer, $\boldsymbol{W}_{L+1} \in \mathbb{R}^{d_{L+1} \times d_L}$ the weight matrix between the last hidden layer and the output layer, $\boldsymbol{b}_{L+1} \in \mathbb{R}^{L+1}$ the bias vector of the output layer. We denote a general activation function by $h$, $\mathcal{W} := \{\boldsymbol{W}_\ell\}_{\ell=1}^{L+1}$ and $\boldsymbol{b} := \{\boldsymbol{b}_\ell\}_{\ell=1}^{L+1}$.

### 3.2 PROBLEM FORMULATION

In the training of regularized DNNs, we are solving the following optimization problem:

$$\min_{\boldsymbol{a},\boldsymbol{z},\mathcal{W},\boldsymbol{b}} F(\boldsymbol{a},\mathcal{W},\boldsymbol{b}) \equiv \gamma_{L+1} \sum_{j=1}^{N} \mathcal{L}(\boldsymbol{W}_{L+1}\boldsymbol{a}_{L,j} + \boldsymbol{b}_{L+1}; \boldsymbol{y}_j) + \sum_{\ell=1}^{L+1} r_\ell(\boldsymbol{W}_\ell)$$

$$\text{subject to } \boldsymbol{z}_{\ell,j} = \boldsymbol{W}_\ell \boldsymbol{a}_{\ell-1,j} + \boldsymbol{b}_\ell, \ \boldsymbol{a}_{\ell,j} = h(\boldsymbol{z}_{\ell,j}) \text{ for all } \ell \in \{1,\dots,L\}, j \in \{1,\dots,N\}. \quad (1)$$

where $\mathcal{L}$ is a generic convex loss function and $r_\ell$, $\ell = 1, \dots, L+1$, are convex but possibly nonsmooth regularizers, $\boldsymbol{a} := \{\boldsymbol{a}_{\ell,j}\}_{\ell=1}^{L}$ the set of all activation vectors, $\boldsymbol{z} := \{\boldsymbol{z}_{\ell,j}\}_{\ell=1}^{L}$, $\boldsymbol{a}_{0,j} := \boldsymbol{x}_j$ for all $j \in \{1,\dots,N\}$, and $\gamma_{L+1} > 0$. The problem (1) can be reformulated as follows:

$$\min_{\boldsymbol{a},\boldsymbol{z},\mathcal{W},\boldsymbol{b}} F(\boldsymbol{a},\mathcal{W},\boldsymbol{b}) + \sum_{j=1}^{N}\sum_{\ell=1}^{L} \frac{\rho_\ell}{2}\|h(\boldsymbol{z}_{\ell,j}) - \boldsymbol{a}_{\ell,j}\|^2 + \sum_{j=1}^{N}\sum_{\ell=1}^{L} \frac{\gamma_\ell}{2}\|\boldsymbol{W}_\ell \boldsymbol{a}_{\ell-1,j} + \boldsymbol{b}_\ell - \boldsymbol{z}_\ell\|^2, \quad (2)$$

where $\|\cdot\|$ is the standard Euclidean norm, $\rho_\ell > 0$ and $\gamma_\ell > 0$ for all $\ell \in \{1,\dots,L\}$. The above scheme allows for any general activation functions such as hyperbolic tangent, sigmoid and ReLU. However, the formulation (2) is generally hard to solve explicitly (say, for tanh and sigmoid) but can be simplified if we consider the constraint $\boldsymbol{a}_{\ell,j} = h(\boldsymbol{z}_{\ell,j})$ as a projection onto a convex set. For instance, ReLU can be thought of as a projection onto the closed upper half-space. This is equivalent to imposing the constraint $\boldsymbol{a}_{\ell,j} \succeq \boldsymbol{0} \Leftrightarrow \boldsymbol{a}_{\ell,j} \in \mathbb{R}_+^{d_\ell}$ for all $\ell \in \{1,\dots,L\}$, and for all $j \in \{1,\dots,N\}$. For hyperbolic tangent and sigmoid functions, nonsmooth approximations are needed and are discussed in Appendix A.1. Inspired by the formulation in Zhang et al. (2016), we further introduce a set of auxiliary variables to the objective to get:

$$\min_{\boldsymbol{a},\mathcal{W},\boldsymbol{b},\boldsymbol{u}} \widetilde{F}(\boldsymbol{a},\mathcal{W},\boldsymbol{b},\boldsymbol{u}) \equiv F(\boldsymbol{a},\mathcal{W},\boldsymbol{b}) + \sum_{j=1}^{N}\sum_{\ell=1}^{L} \left[ \frac{\gamma_\ell}{2}\|\boldsymbol{W}_\ell \boldsymbol{a}_{\ell-1,j} + \boldsymbol{b}_\ell - \boldsymbol{a}_{\ell,j} + \boldsymbol{u}_{\ell,j}\|^2 + \iota_{\mathcal{S}_\ell}(\boldsymbol{a}_{\ell,j}) \right],$$

$$(3)$$

where $\boldsymbol{u} := \{\boldsymbol{u}_{\ell,j}\}_{\ell=1}^{L}$ is the set of auxiliary variables, $\mathcal{S}_\ell$ is a nonempty closed convex set and $\iota_{\mathcal{S}_\ell}$ is the indicator function of a nonempty closed convex set $\mathcal{S}_\ell$ so that $\iota_{\mathcal{S}_\ell}(\boldsymbol{u}) = 0$ for $\boldsymbol{u} \in \mathcal{S}_\ell$ and $+\infty$ otherwise. For the case of the ReLU activation function, $\mathcal{S}_\ell = \mathbb{R}_+^{d_\ell}$. This formulation (3) is more desirable since we eliminate the variable $\boldsymbol{z}$ to be optimized which probably speeds up the training. Also note that the objective function (3) is block multiconvex which allows for established convergence guarantees in existing literature using the proposed algorithm.

### 3.3 THE PROPOSED ALGORITHM

In our minimization algorithm, we perform a proximal step (as in the proximal point algorithm) for each parameter except the auxiliary variables (which are updated by direct minimization instead of dual gradient ascent in Zhang et al. (2016)) in a Gauss-Seidel fashion, and in a backward order based on the network structure. Adaptive momentum (Lau & Yao, 2017), though not included in the convergence analysis, is also used after each proximal point update due to its empirical usefulness for convergence. The overall algorithm is depicted in Algorithm 1 (see Appendix A.2).

### 3.4 CONVERGENCE RESULTS

The problem formulation (3) involves regularized block multiconvex optimization and the proposed algorithm fits the general framework of Xu & Yin (2013). We analyze the convergence of Algorithm 1 under the assumptions in Assumption 1 (see Appendix A.4). The proof of the following theorem and its related convergence rate is given in Appendix A.4.

**Theorem 1 (Global convergence)** *Under Assumption 1, Proposition 1 and the fact that the sequence $\{\boldsymbol{x}^{(k)}\}_{k \geq 1} := \{\boldsymbol{a}^{(k)}, \mathcal{W}^{(k)}, \boldsymbol{b}^{(k)}, \boldsymbol{u}^{(k)}\}_{k \geq 1}$ generated by Algorithm 1 has a finite limit point $\bar{\boldsymbol{x}} := \{\bar{\boldsymbol{a}}, \overline{\mathcal{W}}, \bar{\boldsymbol{b}}, \bar{\boldsymbol{u}}\}$ where $\widetilde{F}$ satisfies the the Kurdyka-Łojasiewicz (KL) property (Definition 2, see Appendix A.4), the sequence $\{\boldsymbol{x}^{(k)}\}_{k \geq 1}$ converges to $\bar{\boldsymbol{x}}$, which is a critical point of* (3).

## 4 EXPERIMENTAL RESULTS

We conduct experiments for two different structures on CIFAR-10 (Krizhevsky et al., 2009) with 50K training and 10K test samples, namely a 3072-4K-4K-4K-10 MLP and a 3072-4K-3072-4K-10 DNN with a residual connection in the second hidden layer (ResNet (He et al., 2016)). Experimental results on MNIST are in Appendix B. The BCD algorithm (20 epochs) is implemented using MATLAB while backprop (SGD; 100 epochs) is implemented using Keras with TensorFlow backend. Squared losses, ReLUs are used without regularizations. All weight matrices are initialized from a Gaussian distribution with a standard deviation of 0.01 and the bias vectors are initialized as vectors of all 0.1, while $\boldsymbol{a}$ and $\boldsymbol{u}$ are initialized by a single forward pass. Hyperparameters in BCD ($\beta_i = 0.95, \gamma_i = 0.1, t = 0.1, s = 1$ (MLP), $s = 0.1$ (ResNet)) and the learning rate (0.05) in SGD are tuned manually. We report the training and test accuracies (the median of 5 runs) as follows:

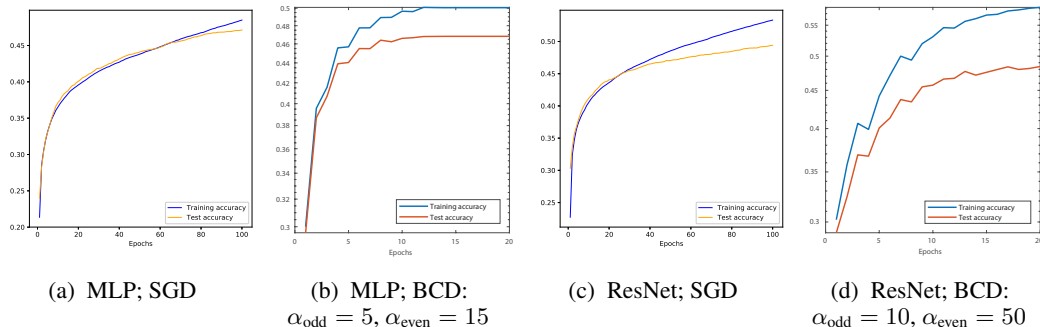

(a) MLP; SGD    (b) MLP; BCD: $\alpha_{\text{odd}} = 5, \alpha_{\text{even}} = 15$    (c) ResNet; SGD    (d) ResNet; BCD: $\alpha_{\text{odd}} = 10, \alpha_{\text{even}} = 50$

Figure 1: Training and test accuracies. Final test acc.: 1a: 0.4765; 1b: 0.4682; 1c: 0.494; 1d: 0.4843.

## 5 CONCLUSION AND FUTURE WORK

In this paper, we proposed an efficient BCD algorithm and established its convergence guarantees according to our block multiconvex formulation. Three major advantages of BCD include: (a) high per epoch efficiency at early stages (observed in Figure 1), i.e., the training and test accuracies of BCD grow much faster than SGD in terms of epoch at the early stage; (b) good scalability, i.e., BCD can be implemented in a distributed and parallel manner via data parallelism on multi-core CPUs; (c) gradient free, i.e., gradient computations are unnecessary used for the updates. One flaw of the BCD methods is that they generally require more memory than SGD method. Thus, a future direction is to study the feasibility of the stochastic and parallel block coordinate descent methods for DNN training.

ACKNOWLEDGMENTS

The authors would like to thank Ruohan Zhan for sharing experimental results on deep neural network training using BCD and ADMM algorithms.

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

## APPENDICES

## A  ALGORITHMIC DETAILS

### A.1  SIMPLIFICATIONS OF ACTIVATION FUNCTIONS

We first give the definition of the proximity operator which is required in the following analysis.

**Definition 1 (Proximity operator (Moreau, 1962; Combettes & Pesquet, 2011))**
Let $\lambda$ be a positive parameter, $\mathcal{H}$ be a real Hilbert space (e.g., Euclidean space) and the function $g : \mathcal{H} \to (-\infty, +\infty]$. The *proximity operator* $\text{prox}_{\lambda g} : \mathcal{H} \to \mathcal{H}$ of the function $\lambda g$ is defined through

$$\text{prox}_{\lambda g}(x) := \underset{y \in \mathcal{H}}{\text{argmin}}\, g(y) + \frac{1}{2\lambda}\|y - x\|^2.$$

If $g$ is convex, proper and lower semicontinuous, $\text{prox}_g$ admits a unique solution. If $g$ is nonconvex, then it is generally set-valued.

For the ReLU activation function, we can consider it as the projection onto the nonempty closed convex set $\mathbb{R}^{d_\ell}_+$, where $d_\ell$ is the dimension of the input variable. This is based on Bauschke & Combettes (2017, Proposition 24.47), which states that for any proper lower semicontinuous convex function $\phi$ on $\mathbb{R}$ and any closed interval $\Omega$ on $\mathbb{R}$ such that $\Omega \cap \text{dom}\, \phi \neq \emptyset$,

$$\text{prox}_{\phi + \iota_\Omega} = \mathcal{P}_\Omega \circ \text{prox}_\phi,$$

where $\mathcal{P}_\Omega \equiv \text{prox}_{\iota_\Omega}$ is the projection operator onto the nonempty closed convex set $\Omega$. Since all the activation functions are elementwise operations, according to Bauschke & Combettes (2017, Proposition 24.11), we can extend the above results to the Euclidean space $\mathbb{R}^{d_\ell}$, i.e.,

$$(\forall \boldsymbol{x} \in \mathbb{R}^{d_\ell}) \quad \text{prox}_{\boldsymbol{\phi} + \iota_{\boldsymbol{\Omega}}}(\boldsymbol{x}) = \left(\text{prox}_{\phi_i + \iota_{\Omega_i}}(x_i)\right)_{1 \leq i \leq d_\ell} = \left(\mathcal{P}_{\Omega_i} \circ \text{prox}_{\phi_i}(x_i)\right)_{1 \leq i \leq d_\ell},$$

where $\boldsymbol{x} = (x_i)_{1 \leq i \leq d_\ell}$, $\boldsymbol{\phi} = \bigoplus_{i=1}^{d_\ell} \phi_i$ and $\boldsymbol{\Omega} = \prod_{i=1}^{d_\ell} \Omega_i$.

Likewise, the sigmoid and tanh activation functions can be used with some tricks in this scheme. It should be noted that these two functions are not simple projections onto nonempty closed convex sets. Instead, if we consider the nonsmooth surrogates of them, they can be imposed as projections onto nonempty closed convex sets which are much easier to obtain.

For the tanh function, we use the following nonsmooth function (a.k.a. *hard tanh*) as an approximation:

$$f(x) = \begin{cases} -1 & \text{if } x < -1, \\ x & \text{if } x \in [-1, 1], \\ 1 & \text{if } x > 1. \end{cases}$$

Then we have

$$\mathcal{P}_{[-1,1]}(x) = \text{prox}_{\iota_{[-1,1]}}(x) = \max\{-1, \min\{x, 1\}\}.$$

For the sigmoid function $\sigma$, recall that we have the following relationship:

$$\sigma(x) \equiv 0.5(1 + \tanh(0.5x)).$$

Using function transformation, the nonsmooth approximation of the sigmoid function (a.k.a. *hard sigmoid*[1]) is

$$g(x) = \begin{cases} 0 & \text{if } x < -2, \\ 0.25x + 0.5 & \text{if } x \in [-2, 2], \\ 1 & \text{if } x > 2. \end{cases}$$

We define the closed convex set $\Sigma := \{x \in \mathbb{R} : u \in [-2, 2], x = 0.25u + 0.5\}$. Then we have

$$\mathcal{P}_\Sigma(x) = \text{prox}_{\iota_\Sigma}(x) = \max\{0, \min\{0.25x + 0.5, 1\}\} = 0.25 \max\{-2, \min\{x, 2\}\} + 0.5.$$

---

[1]Hard sigmoid can also be defined as: $g(x) = 0$ if $x < 2.5$, $g(x) = 0.2x + 0.5$ if $x \in [-2.5, 2.5]$, and $g(x) = 1$ if $x > 2.5$, as defined in TensorFlow.

## A.2 THE PROXIMAL BCD ALGORITHM

Note that the extrapolations and adaptive momentums in the following algorithm are not considered in the convergence results but implemented in numerical experiments.

---

**Algorithm 1:** Proximal Block Coordinate Descent (BCD) Algorithm for Training DNNs

---

**Input:** training data $\{(\boldsymbol{x}_j, \boldsymbol{y}_j)\}_{j=1}^N$ and regularization parameters $\{\gamma_\ell\}_{\ell=1}^{L+1}$ and $\boldsymbol{a}_{L+1,j}^{(k)} := \boldsymbol{y}_j$ for all $k \in \mathbb{N}$ and $j \in \{1, \dots, N\}$.

Initialize $\mathcal{W}^{(0)}, \boldsymbol{b}^{(0)}, \alpha_i^{(0)} \in (0, \infty), \beta_i^{(0)} \in (0,1)$ for all $i$, and $s, t \in (0, 1]$; $\boldsymbol{a}^{(0)}$ initialized by forward propagation.

**for** $k = 1, 2, \dots$ **do**

$\quad W_{L+1}^* = \operatorname{argmin}_{W_{L+1}} \gamma_{L+1} \mathcal{L}(W_{L+1} \boldsymbol{a}_{L,j}^{(k-1)} + \boldsymbol{b}_{L+1}^{(k-1)}; \boldsymbol{y}_j) + \frac{\alpha_1^{(k-1)}}{2} \| W_{L+1} -$
$\quad W_{L+1}^{(k-1)} \|_F^2 + r_{L+1}(W_{L+1})$

$\quad W_{L+1}^{(k)} = W_{L+1}^{(k-1)} + \beta_1^{(k-1)}(W_{L+1}^* - W_{L+1}^{(k-1)})$

$\quad \boldsymbol{b}_{L+1}^* = \operatorname{argmin}_{\boldsymbol{b}_{L+1}} \gamma_{L+1} \mathcal{L}(W_{L+1}^{(k-1)} \boldsymbol{a}_{L,j}^{(k-1)} + \boldsymbol{b}_{L+1}; \boldsymbol{y}_j) + \frac{\alpha_1^{(k-1)}}{2} \| \boldsymbol{b}_{L+1} - \boldsymbol{b}_{L+1}^{(k-1)} \|^2$

$\quad \boldsymbol{b}_{L+1}^{(k)} = \boldsymbol{b}_{L+1}^{(k-1)} + \beta_1^{(k-1)}(\boldsymbol{b}_{L+1}^* - \boldsymbol{b}_{L+1}^{(k-1)})$

$\quad$ **if** $F(\cdots, W_{L+1}^*, \boldsymbol{b}_{L+1}^*, \cdots) \le F(\cdots, W_{L+1}^{(k)}, \boldsymbol{b}_{L+1}^{(k)}, \cdots)$ **then**
$\quad\quad \lfloor \; \beta_1^{(k)} = t\beta_1^{(k-1)}$

$\quad$ **else**
$\quad\quad \lfloor \; \beta_1^{(k)} = \min\{\beta_1^{(k-1)}/s, 1\}$

$\quad$ **for** $\ell = L, \dots, 1$ **do**

$\quad\quad \boldsymbol{a}_{\ell,j}^* = \operatorname{argmin}_{\boldsymbol{a}_{\ell,j}} \frac{\gamma_{\ell+1}}{2} \| W_{\ell+1}^{(k)} \boldsymbol{a}_{\ell,j} + \boldsymbol{b}_{\ell+1}^{(k)} - \boldsymbol{a}_{\ell+1,j}^{(k)} + \boldsymbol{u}_{\ell+1,j}^{(k)} \|^2 +$
$\quad\quad \frac{\gamma_\ell}{2} \| W_\ell^{(k-1)} \boldsymbol{a}_{\ell-1,j}^{(k-1)} + \boldsymbol{b}_\ell^{(k-1)} - \boldsymbol{a}_{\ell,j} + \boldsymbol{u}_{\ell,j}^{(k)} \|^2 + \frac{\alpha_{2(L-\ell+1)}^{(k-1)}}{2} \| \boldsymbol{a}_{\ell,j} - \boldsymbol{a}_{\ell,j}^{(k-1)} \|^2 + \iota_{\mathcal{S}_\ell}(\boldsymbol{a}_{\ell,j})$
$\quad\quad$ for all $j \in \{1, \dots, N\}$

$\quad\quad \boldsymbol{a}_{\ell,j}^{(k)} = \boldsymbol{a}_{\ell,j}^{(k-1)} + \beta_{2(L-\ell+1)}^{(k-1)}(\boldsymbol{a}_{\ell,j}^* - \boldsymbol{a}_{\ell,j}^{(k-1)})$ for all $j \in \{1, \dots, N\}$

$\quad\quad$ **if** $F(\cdots, \boldsymbol{a}_{\ell,1}^*, \dots, \boldsymbol{a}_{\ell,N}^*, \cdots) \le F(\cdots, \boldsymbol{a}_{\ell,1}^{(k)}, \dots, \boldsymbol{a}_{\ell,N}^{(k)}, \cdots)$ **then**
$\quad\quad\quad \lfloor \; \beta_{2(L-\ell+1)}^{(k)} = t\beta_{2(L-\ell+1)}^{(k-1)}$

$\quad\quad$ **else**
$\quad\quad\quad \lfloor \; \beta_{2(L-\ell+1)}^{(k)} = \min\{\beta_{2(L-\ell+1)}^{(k-1)}/s, 1\}$

$\quad\quad \boldsymbol{u}_{\ell,j}^{(k)} = \operatorname{argmin}_{\boldsymbol{u}_{\ell,j}} \frac{\gamma_\ell}{2} \| W_\ell^{(k-1)} \boldsymbol{a}_{\ell-1,j}^{(k-1)} + \boldsymbol{b}_\ell^{(k-1)} - \boldsymbol{a}_{\ell,j}^{(k)} + \boldsymbol{u}_{\ell,j} \|^2$

$\quad\quad W_\ell^* = \operatorname{argmin}_{W_\ell} \sum_{j=1}^N \frac{\gamma_\ell}{2} \| W_\ell \boldsymbol{a}_{\ell-1,j}^{(k-1)} + \boldsymbol{b}_\ell^{(k-1)} - \boldsymbol{a}_{\ell,j}^{(k)} + \boldsymbol{u}_{\ell,j}^{(k)} \|^2 +$
$\quad\quad \frac{\alpha_{2(L-\ell+1)+1}^{(k-1)}}{2} \| W_\ell - W_\ell^{(k-1)} \|_F^2 + r_\ell(W_\ell)$

$\quad\quad W_\ell^{(k)} = W_\ell^{(k-1)} + \beta_{2(L-\ell+1)+1}^{(k)}(W_\ell^* - W_\ell^{(k-1)})$

$\quad\quad \boldsymbol{b}_\ell^* =$
$\quad\quad \operatorname{argmin}_{\boldsymbol{b}_\ell} \sum_{j=1}^N \frac{\gamma_\ell}{2} \| W_\ell^{(k-1)} \boldsymbol{a}_{\ell-1,j}^{(k-1)} + \boldsymbol{b}_\ell - \boldsymbol{a}_{\ell,j}^{(k)} + \boldsymbol{u}_{\ell,j}^{(k)} \|^2 + \frac{\alpha_{2(L-\ell+1)+1}^{(k-1)}}{2} \| \boldsymbol{b}_\ell - \boldsymbol{b}_\ell^{(k-1)} \|^2$

$\quad\quad \boldsymbol{b}_\ell^{(k)} = \boldsymbol{b}_\ell^{(k-1)} + \beta_{2(L-\ell+1)+1}^{(k-1)}(\boldsymbol{b}_\ell^* - \boldsymbol{b}_\ell^{(k-1)})$

$\quad\quad$ **if** $F(\cdots, W_\ell^*, \boldsymbol{b}_\ell^*, \cdots) \le F(\cdots, W_\ell^{(k)}, \boldsymbol{b}_\ell^{(k)}, \cdots)$ **then**
$\quad\quad\quad \lfloor \; \beta_{2(L-\ell+1)+1}^{(k)} = t\beta_{2(L-\ell+1)+1}^{(k-1)}$

$\quad\quad$ **else**
$\quad\quad\quad \lfloor \; \beta_{2(L-\ell+1)+1}^{(k)} = \min\{\beta_{2(L-\ell+1)+1}^{(k-1)}/s, 1\}$

**Output:** $\mathcal{W}, \boldsymbol{b}$

---

## A.3 IMPLEMENTATION DETAILS

For instance, if we take $r(\boldsymbol{W}_\ell) \equiv \lambda \|\boldsymbol{W}_\ell\|_F^2$, then we have

$$
\boldsymbol{W}_{L+1}^* = \left( \alpha_1^{(k-1)} \boldsymbol{W}_{L+1}^{(k-1)} + \gamma_{L+1} \sum_{j=1}^N (\boldsymbol{y}_j - \boldsymbol{b}_{L+1}^{(k-1)})(\boldsymbol{a}_{L,j}^{(k-1)})^\top \right)
$$

$$
\left( (\alpha_1^{(k-1)} + \lambda) \boldsymbol{I}_{d_L} + \gamma_{L+1} \sum_{j=1}^N \boldsymbol{a}_{L,j}^{(k-1)} (\boldsymbol{a}_{L,j}^{(k-1)})^\top \right)^{-1}
$$

$$
\boldsymbol{b}_{L+1}^* = \frac{1}{1 + \alpha_1^{(k-1)}} \left( \alpha_1^{(k-1)} \boldsymbol{b}_{L+1}^{(k-1)} + \gamma_{L+1} \sum_{j=1}^N (\boldsymbol{y}_j - \boldsymbol{W}_{L+1}^{(k-1)} \boldsymbol{a}_{L,j}^{(k-1)}) \right)
$$

For all $j \in \{1, \ldots, N\}$, and for $\ell = L, \ldots, 1$,

$$
\boldsymbol{a}_{\ell,j}^* = \left( \gamma_{\ell+1} \boldsymbol{W}_{\ell+1}^{(k)} (\boldsymbol{W}_{\ell+1}^{(k)})^\top + (\alpha_{2(L-\ell+1)}^{(k-1)} + \gamma_\ell) \boldsymbol{I}_{d_\ell} \right)^{-1} \left( \gamma_{\ell+1} (\boldsymbol{W}_{\ell+1}^{(k)})^\top (\boldsymbol{a}_{\ell+1,j}^{(k)} - \boldsymbol{b}_{\ell+1}^{(k)}) \right.
$$

$$
\left. + \gamma_\ell (\boldsymbol{W}_\ell^{(k-1)} \boldsymbol{a}_{\ell-1,j}^{(k-1)} + \boldsymbol{b}_\ell^{(k-1)} + \boldsymbol{u}_{\ell,j}^{(k-1)}) + \alpha_{2(L-\ell+1)}^{(k-1)} \boldsymbol{a}_{\ell,j}^{(k-1)} \right)
$$

$$
\boldsymbol{a}_{\ell,j}^* = \mathcal{P}_{\mathcal{S}_\ell}(\boldsymbol{a}_{\ell,j}^*)
$$

$$
\boldsymbol{u}_{\ell,j}^* = \boldsymbol{a}_{\ell,j}^{(k)} - \boldsymbol{W}_\ell^{(k-1)} \boldsymbol{a}_{\ell-1,j}^{(k-1)} - \boldsymbol{b}_\ell^{(k-1)}
$$

$$
\boldsymbol{W}_\ell^* = \left( \alpha_{2(L-\ell+1)+1}^{(k-1)} \boldsymbol{W}_\ell^{(k-1)} + \gamma_\ell \sum_{j=1}^N (\boldsymbol{a}_{\ell,j}^{(k)} - \boldsymbol{b}_\ell^{(k-1)} - \boldsymbol{u}_{\ell,j}^{(k)})(\boldsymbol{a}_{\ell-1,j}^{(k-1)})^\top \right)
$$

$$
\left( \alpha_{2(L-\ell+1)+1}^{(k-1)} \boldsymbol{I}_{d_{\ell-1}} + \gamma_\ell \sum_{j=1}^N \boldsymbol{a}_{\ell-1,j}^{(k-1)} (\boldsymbol{a}_{\ell-1,j}^{(k-1)})^\top \right)^{-1}
$$

$$
\boldsymbol{b}_\ell^* = \frac{1}{\gamma_\ell N + \alpha_{2(L-\ell+1)+1}^{(k-1)}} \left( \alpha_{2(L-\ell+1)+1}^{(k-1)} \boldsymbol{b}_\ell^{(k-1)} + \gamma_\ell \sum_{j=1}^N (\boldsymbol{a}_{\ell,j}^{(k)} - \boldsymbol{W}_\ell^{(k-1)} \boldsymbol{a}_{\ell-1,j}^{(k-1)}) \right)
$$

## A.4 ASSUMPTIONS, DEFINITIONS, PROPOSITIONS AND RELATED PROOFS

**Assumption 1** *We have several assumptions on the functions $\widetilde{F}$: (i) The loss function $\mathcal{L}$ is chosen such that $\widetilde{F}$ is continuous[2] and bounded below on $\operatorname{dom} \widetilde{F}$. Problem (3) has a Nash point (Xu & Yin, 2013); (ii) For each block $i$, there exist constant $0 < a_i \leq A_i < \infty$ such that the proximal parameters $\alpha_i^{(k-1)}$ obeys $a_i \leq \alpha_i^{(k-1)} \leq A_i$.*

**Definition 2 (Kurdyka-Łojasiewicz property and KL function (Bolte et al., 2014))**

1. The function $F : \mathbb{R}^n \to \mathbb{R} \cup \{+\infty\}$ is said to have the *Kurdyka-Łojasiewicz property* at $\bar{\boldsymbol{x}} \in \operatorname{dom} \partial F$ if there exist $\eta \in (0, +\infty]$, a neighborhood $\mathcal{B}_\rho(\bar{\boldsymbol{x}}) := \{\boldsymbol{x} : \|\boldsymbol{x} - \bar{\boldsymbol{x}}\| < \rho\}$ of $\bar{\boldsymbol{x}}$ and a continuous concave function $\varphi(t) := ct^{1-\theta}$ for some $c > 0$ and $\theta \in [0, 1)$ such that for all $\boldsymbol{x} \in \mathcal{B}_\rho(\bar{\boldsymbol{x}}) \cap [F(\bar{\boldsymbol{x}}) < F < F(\bar{\boldsymbol{x}}) + \eta]$, the Kurdyka-Łojasiewicz inequality holds

$$
\varphi'(F(\boldsymbol{x}) - F(\bar{\boldsymbol{x}})) \operatorname{dist}(\boldsymbol{0}, \partial F(\boldsymbol{x})) \geq 1.
$$

---

[2] Note that the indicator function $\iota_{\mathcal{S}_\ell}$ is continuous on $\operatorname{dom}(\iota_{\mathcal{S}_\ell})$ since according to its definition, $\iota_{\mathcal{S}_\ell}(\boldsymbol{a}_{\ell,j}) = 0$ if $\boldsymbol{a}_{\ell,j} \in \operatorname{dom}(\iota_{\mathcal{S}_\ell}) \equiv \mathbb{R}_+^{d_\ell}$ and $\iota_{\mathcal{S}_\ell}(\boldsymbol{a}_{\ell,j}) = \infty$ otherwise, for all $j \in \{1, \ldots, N\}$ and $\ell \in \{1, \ldots, L\}$. In general, the indicator function $\iota$ is called an *extended-value* convex function since it equals $+\infty$ if the variable is not in the domain of $\iota$.

2. Proper lower semincontinuous functions which satisfy the Kurdyka-Łojasiewicz inequality at each point of $\mathrm{dom}\,\partial F$ are called *KL functions*.

**Definition 3 (Semialgebraic function)**

1. A set $\mathcal{D} \subset \mathbb{R}^n$ is called semialgebraic (Bochnak et al., 1998) if it can be represented as

$$\mathcal{D} = \bigcup_{i=1}^{s} \bigcap_{j=1}^{t} \{\boldsymbol{x} \in \mathbb{R}^n : p_{ij}(\boldsymbol{x}) = 0, q_{ij}(\boldsymbol{x}) > 0\},$$

where $p_{ij}, q_{ij}$ are real polynomial functions for $1 \le i \le s, 1 \le j \le t$.

2. A function $h$ is called *semialgebraic* if its graph $\mathrm{Gr}(h) := \{(x, h(x)) : x \in \mathrm{dom}(h)\}$ is a semialgebraic set.

**Remark 1** KL functions include real analytic functions, functions on the *o*-minimal structure (Kurdyka, 1998), subanalytic functions (Bolte et al., 2007a;b), semialgebraic functions (see Definition 3) and locally strongly convex functions. Some other important facts regarding KL functions available in Łojasiewicz (1963; 1993); Kurdyka (1998); Bolte et al. (2007a;b); Attouch et al. (2013); Xu & Yin (2013) and references therein are summarized below.

1. The sum of a real analytic function and a semialgebraic function is a subanalytic function, and thus a KL function.

2. If a set $\mathcal{D}$ is semialgebraic, so is its closure $\mathrm{cl}(\mathcal{D})$.

3. If $\mathcal{D}_1$ and $\mathcal{D}_2$ are both semialgebraic, so are $\mathcal{D}_1 \cup \mathcal{D}_2$, $\mathcal{D}_1 \cap \mathcal{D}_2$, and $\mathbb{R}^n \setminus \mathcal{D}_1$.

4. Indicator functions of semialgebraic sets are semialgebraic, e.g., the indicator functions of nonnegative closed half space and a nonempty closed interval.

5. Finite sums and products of semialgebraic (real analytic) functions are semialgebraic (real analytic).

6. The composition of semialgebraic functions is semialgebraic.

**Proposition 1** *The objective function $\widetilde{F}$ in (3) is a KL function, if the loss function $\mathcal{L}$ is chosen as one of the commonly used loss functions such as the squared loss, logistic loss, hinge loss or softmax cross-entropy loss, and the regularizers $r_\ell$'s are chosen as $\ell_1$ norms, squared $\ell_2$ norms (a.k.a. weight decay), $\ell_q$ quasi-norms with $q \in [0, 1)$, or their sums such as the elastic net (Zou & Hastie, 2005).*

PROOF Recall that

$$\widetilde{F}(\boldsymbol{a}, \mathcal{W}, \boldsymbol{b}, \boldsymbol{u}) \equiv \underbrace{\gamma_{L+1} \sum_{j=1}^{N} \mathcal{L}(\boldsymbol{W}_{L+1}\boldsymbol{a}_{L,j} + \boldsymbol{b}_{L+1}; \boldsymbol{y}_j)}_{\widetilde{F}_1(\mathcal{W}, \boldsymbol{b})} + \underbrace{\sum_{\ell=1}^{L+1} r_\ell(\boldsymbol{W}_\ell)}_{\widetilde{F}_2(\mathcal{W})}$$

$$+ \underbrace{\sum_{j=1}^{N} \sum_{\ell=1}^{L} \frac{\gamma_\ell}{2} \|\boldsymbol{W}_\ell \boldsymbol{a}_{\ell-1,j} + \boldsymbol{b}_\ell - \boldsymbol{a}_{\ell,j} + \boldsymbol{u}_{\ell,j}\|^2}_{\widetilde{F}_3(\boldsymbol{a}, \mathcal{W}, \boldsymbol{b}, \boldsymbol{u})} + \underbrace{\sum_{j=1}^{N} \sum_{\ell=1}^{L} \iota_{\mathcal{S}_\ell}(\boldsymbol{a}_{\ell,j})}_{\widetilde{F}_4(\boldsymbol{a})}.$$

Any polynomial function is real analytic, so $\widetilde{F}_3$ is real analytic by Remark 1 item 5. In the same vein, the square loss function is also real analytic. The logistic loss and softmax cross-entropy loss are also real analytic (Xu & Yin, 2013). If $\mathcal{L}$ is the hinge loss, i.e., given $\boldsymbol{y} \in \mathbb{R}^{d_{L+1}}$, $\mathcal{L}(\boldsymbol{u}, \boldsymbol{y}) := \max\{0, 1 - \boldsymbol{u}^\top \boldsymbol{y}\}$ for any $\boldsymbol{u} \in \mathbb{R}^{d_{L+1}}$, it is semialgebraic, because its graph is $\mathrm{cl}(\mathcal{D})$, a closure of the set $\mathcal{D}$, where

$$\mathcal{D} = \{(\boldsymbol{u}, z) : 1 - \boldsymbol{u}^\top \boldsymbol{y} - z = 0, \boldsymbol{1} - \boldsymbol{u} \succ 0\} \cup \{(\boldsymbol{u}, z) : z = 0, \boldsymbol{u}^\top \boldsymbol{y} - 1 > 0\}.$$

Then again by Remark 1 item 5, $\widetilde{F}_1$ is either a real analytic function (squared, logistic and softmax cross-entropy losses) or a semialgebraic function (hinge loss). $\widetilde{F}_4$ is semialgebraic by Remark 1 items 4 and 5 since $\mathcal{S}_\ell$ is a nonempty closed interval for all $\ell \in \{1, \ldots, L\}$ depicted in Appendix A.1.

Concerning $\widetilde{F}_2$, which is the sum of the regularizers $r_\ell$'s, note that the $\ell_1$ norm, the squared $\ell_2$ norm, the $\ell_q$ quasi-norms with $q \in [0, 1)$ are all semialgebraic, and thus, the elastic net is also semialgebraic. By Remark 1 item 5, $\widetilde{F}_2$ is also semialgebraic.

Finally, using Remark 1 item 1, we conclude that $\widetilde{F}$ is subanalytic and hence a KL function. ∎

We now present the proof of Theorem 1.

PROOF (THEOREM 1) Note that $\widetilde{F}$ is monotonically nonincreasing and converges to $\widetilde{F}(\bar{x})$. If $\widetilde{F}(x^{(k_0)}) = \widetilde{F}(\bar{x})$ at some $k_0$, then $x^{(k)} = x^{(k_0)} = \bar{x}$ for all $k \geq k_0$. It remains to consider $\widetilde{F}(x^{(k)}) > \widetilde{F}(\bar{x})$ for all $k \geq 0$. Since $\bar{x}$ is a limit point and $\widetilde{F}(x^{(k)}) \to \widetilde{F}(\bar{x})$, there must exist an integer $k_0$ such that $x^{(k_0)}$ is sufficiently close to $\bar{x}$. Hence, $\{x^{(k)}\}_{k \geq 1}$ converges according to Xu & Yin (2013, Lemma 2.6). ∎

**Theorem 2 (Convergence rate)** *Suppose that* $\{x^{(k)}\}_{k \geq 1} := \{a^{(k)}, \mathcal{W}^{(k)}, b^{(k)}, u^{(k)}\}_{k \geq 1}$ *converges to a critical point* $\bar{x} := \{\bar{a}, \overline{\mathcal{W}}, \bar{b}, \bar{u}\}$, *at which* $\widetilde{F}$ *satisfies the KL property with* $\varphi(t) := ct^{1-\theta}$ *for some* $c > 0$ *and* $\theta \in [0, 1)$. *Then the following hold:*

1. *If* $\theta = 0$, $x^{(k)}$ *converges to* $\bar{x}$ *in finitely many iterations.*

2. *If* $\theta \in (0, 1/2]$, $\|x^{(k)} - \bar{x}\| \leq C\tau^k$ *for all* $k \geq k_0$, *for certain* $k_0 > 0$, $C > 0$, $\tau \in [0, 1)$.

3. *If* $\theta \in (1/2, 1)$, $\|x^{(k)} - \bar{x}\| \leq Ck^{-(1-\theta)/(2\theta-1)}$ *for all* $k \geq k_0$, *for certain* $k_0 > 0$, $C > 0$.

*These three parts correspond to finite convergence, linear convergence, and sublinear convergence, respectively.*

PROOF See the proof of Theorem 2.9 of Xu & Yin (2013). ∎

## B  FURTHER EXPERIMENTAL RESULTS

We further conduct experiments for two different structures on the MNIST dataset (LeCun & Cortes, 2010) with 60K training and 10K test samples, namely a 784-2048-2048-2048-10 MLP and a 784-2048-784-2048-10 DNN with a residual connection in the second hidden layer (ResNet). The BCD algorithm (30 epochs for MLP; 20 epochs for ResNet) is implemented using MATLAB while backprop (SGD; 100 epochs) is implemented using Keras with TensorFlow backend. Squared losses, ReLUs are used without regularizations. All weight matrices are initialized from a Gaussian distribution with a standard deviation of $0.01$ and the bias vectors are initialized as vectors of all $0.1$, while $a$ and $u$ are initialized by a single forward pass. The hyperparameters in BCD ($\beta_i = 0.95, \gamma_i = 0.1, t = 0.1$) and the learning rate ($0.05$) in SGD are chosen by manual tuning. We report the training and test accuracies (the median of 5 runs) as follows:

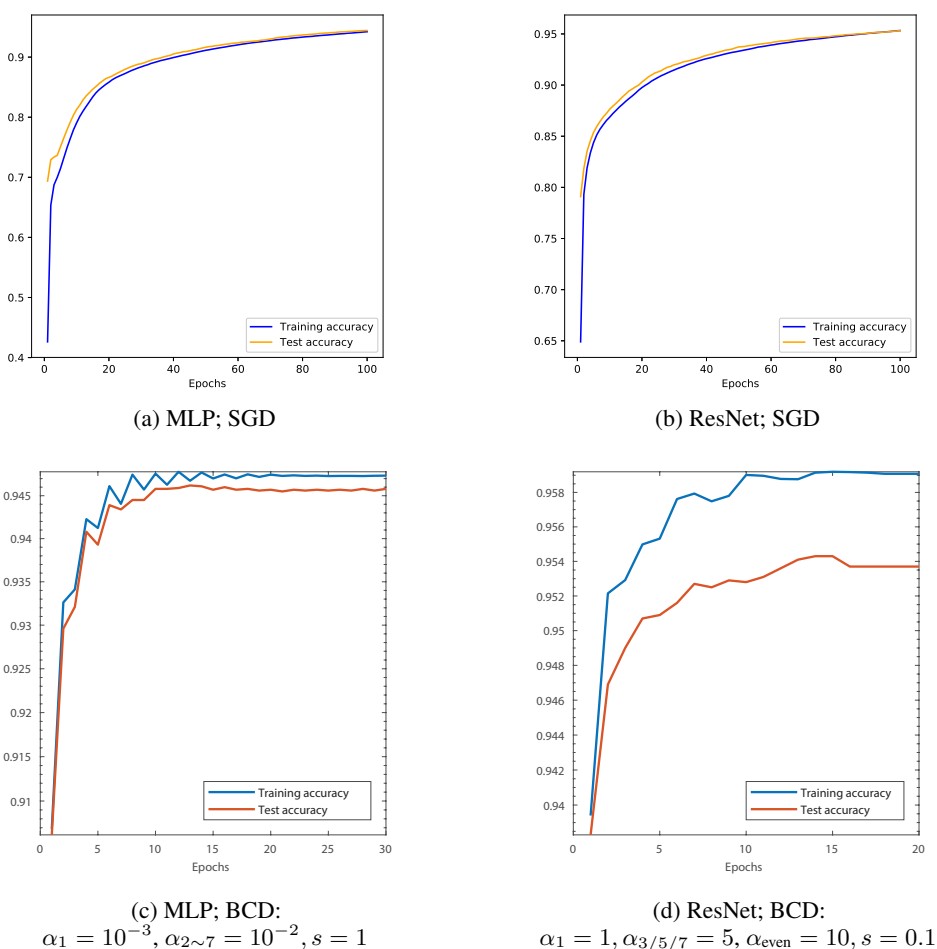

(a) MLP; SGD

(b) ResNet; SGD

(c) MLP; BCD:
$\alpha_1 = 10^{-3}, \alpha_{2 \sim 7} = 10^{-2}, s = 1$

(d) ResNet; BCD:
$\alpha_1 = 1, \alpha_{3/5/7} = 5, \alpha_{\text{even}} = 10, s = 0.1$

Figure 2: Training and test accuracies. Final test acc.: 2a & 2b: 0.9533; 2c: 0.9458; 2d: 0.9537.

From Figure 2, we observe that the BCD algorithms require much fewer epochs to achieve similar test accuracies. Thus, we say that the BCD method has high per epoch efficiency compared to backprop with SGD.

