# OpenReview forum: "A Proximal Block Coordinate Descent Algorithm for Deep Neural Network Training"
_ICLR.cc/2018/Workshop — Accept_

### Official Review · AnonReviewer3 · 2018-03-08
**efficiency of block coordinate descent for training DNNs**

**Rating:** 7
**Confidence:** 1

**Review:**

Training deep neural networks (DNNs) can generally be recast as a nonconvex optimization problem. Many existing methods with guarantees work well for convex cases but rarely for nonconvex cases. This paper exploits the proximal point algorithm combining with the coordinate descent technique to reduce the computational complexity. As far as I see, the numerical experiments look good. However, I am not an expert in this field.

---

### Official Review · AnonReviewer2 · 2018-03-08
**The paper presents a BCD proximal algorithm for minimizing a DNN-type objective. The convergence analysis is based on known results applied to this particular setting where the functions are known to satisfy the KL inequality (e.g. semialgebraic or definable functions).**

**Rating:** 6
**Confidence:** 5

**Review:**

The paper presents a BCD proximal algorithm for minimizing a DNN-type objective. The convergence analysis is based on known results applied to this particular setting where the functions are known to satisfy the KL inequality (e.g. semialgebraic or definable functions). The paper suffers from many issues:
1. The novelty is very limited (essentially applying known convergence results by checking the corresponding assumptions).
2. Key references are missing (e.g.  the work of Sabach, Teboulle, Bolte and many others).
3. Many material is taken from others' work (e.g. material on semialgebraic geometry).
4. Proposition 1 is trivial.
5. Theorem 2 is originally due to the work of Bolte et al. as well as to Frankel et al.
6. The passage from (1) to (2) is NOT rigorous as stated. Seen as a penalty method, for it to work, \gamma_l should be appropriately decreasing. From the Lagrangian "duality" perspective, existing and boundedness of \gamma_l must be proved.
7. In DNN, stochastic versions are applied and it would have been much more interesting to blend the stated analysis with stochastic coordinate descent.

---

### Official Review · AnonReviewer1 · 2018-03-14

**Rating:** 8
**Confidence:** 4

**Review:**

This paper proposes a proximal BCD algorithm for training DNNs, and provide its global convergence results using KL property. My understanding is that the algorithm itself is not very novel. However, its application to DNN is new. I skipped the theoretical analysis in the appendix, and it looks correct and interesting. I thus recommend the paper to be accepted.

I suggest the authors list a pseudo algorithm for the algorithm part for easier presentation, and the comparison between the algorithm and other existing algorithms should be better strengthen.

---

### Decision · Program_Chairs · 2018-03-20
**ICLR 2018 Workshop Acceptance Decision**

**Decision:**

Accept

**Comment:**

Congratulations, your paper was accepted to the ICLR workshop.